# Metabolomic Profile of Different Dietary Patterns and Their Association with Frailty Index in Community-Dwelling Older Men and Women

**DOI:** 10.3390/nu14112237

**Published:** 2022-05-27

**Authors:** Toshiko Tanaka, Sameera A. Talegawkar, Yichen Jin, Julián Candia, Qu Tian, Ruin Moaddel, Eleanor M. Simonsick, Luigi Ferrucci

**Affiliations:** 1Longitudinal Studies Section, National Institute on Aging, Baltimore, MD 21224, USA; julian.candia@nih.gov (J.C.); qu.tian@nih.gov (Q.T.); simonsickel@grc.nia.nih.gov (E.M.S.); ferruccilu@grc.nia.nih.gov (L.F.); 2Department of Exercise and Nutrition Sciences, Milken Institute School of Public Health, The George Washington University, Washington, DC 20052, USA; stalega1@gwu.edu (S.A.T.); yjin@email.gwu.edu (Y.J.); 3Laboratory of Clinical Investigation, National Institute on Aging, Baltimore, MD 21224, USA; moaddelru@grc.nia.nih.gov

**Keywords:** dietary patterns, metabolomics, frailty, mediation, aging

## Abstract

Diet quality has been associated with slower rates of aging; however, the mechanisms underlying the role of a healthy diet in aging are not fully understood. To address this question, we aimed to identify plasma metabolomic biomarkers of dietary patterns and explored whether these metabolites mediate the relationship between diet and healthy aging, as assessed by the frailty index (FI) in 806 participants of the Baltimore Longitudinal Study of Aging. Adherence to different dietary patterns was evaluated using the Mediterranean diet score (MDS), Mediterranean–DASH Diet Intervention for Neurodegenerative Delay (MIND) score, and Alternate Healthy Eating Index-2010 (AHEI). Associations between diet, FI, and metabolites were assessed using linear regression models. Higher adherence to these dietary patterns was associated with lower FI. We found 236, 218, and 278 metabolites associated with the MDS, MIND, and AHEI, respectively, with 127 common metabolites, which included lipids, tri/di-glycerides, lyso/phosphatidylcholine, amino acids, bile acids, ceramides, cholesterol esters, fatty acids and acylcarnitines, indoles, and sphingomyelins. Metabolomic signatures of diet explained 28%, 37%, and 38% of the variance of the MDS, MIND, and AHEI, respectively. Signatures of MIND and AHEI mediated 55% and 61% of the association between each dietary pattern with FI, while the mediating effect of MDS signature was not statistically significant. The high number of metabolites associated with the different dietary patterns supports the notion of common mechanisms that underly the relationship between diet and frailty. The identification of multiple metabolite classes suggests that the effect of diet is complex and not mediated by any specific biomarkers. Furthermore, these metabolites may serve as biomarkers for poor diet quality to identify individuals for targeted dietary interventions.

## 1. Introduction

It has been estimated that between 2015 and 2050, the proportion of individuals aged 60 years or older in the world will double, from 12% to 22% [1]. This substantial increase in the proportion of older adults makes maintenance of functional abilities, both physical and cognitive, in old age an even more compelling objective.

Dietary patterns, defined “as the quantities, proportions, variety, or combination of different foods, drinks, and nutrients in diets, and the frequency with which they are habitually consumed” [2], have been shown to be critical for overall health and well-being and the reduction of age-related functional decline and frailty. A priori or hypothesis-based categorizations of dietary patterns derive from knowledge of food and nutrients, and their associations with health and are used to evaluate adherence to nutritional guidelines and recommendations. In this study, we examined three diet quality summary indices. The Mediterranean diet score (MDS) was developed to assess how closely an individual follows a Mediterranean style diet, which is characterized by a high intake of whole grains, fruits and vegetables, nuts, fish and seafood, and olive oil and a low intake of red meat and dairy foods [3]. The Alternative Healthy Index (AHEI) targets food and nutrients predictive of chronic disease risk and assigns higher scores for foods and nutrients that are considered beneficial for health, including vegetables, fruits and whole grains, and lower scores for sugar-sweetened beverages and juices and red and processed meats, which are considered detrimental [4]. The Mediterranean–DASH Diet Intervention for Neurodegenerative Delay (MIND) dietary pattern evaluates adherence to both a Mediterranean-style diet and the Dietary Approaches to Stop Hypertension (DASH) recommendations [5].

Each of these dietary quality indices has demonstrated associations with aging-related outcomes. For example, in the InCHIANTI study of aging, adherence to a Mediterranean-type diet was inversely associated with developing frailty [6,7], mobility decline, and disability [8]. At the Baltimore Longitudinal Study of Aging (BLSA), participants with higher AHEI scores in middle age had better physical function in older age [8]. Most recently, adherence to the MIND dietary pattern demonstrated protective associations with the maintenance of physical function and muscle strength [9]. While each of these dietary patterns has been associated with better physical and cognitive function in older adults, the underlying biological connection across different dietary assessments and to better functional health in later life remains unknown.

One approach to understanding the biological effects of adherence to dietary patterns is to interrogate the molecular profiles of these patterns in population studies. The identification of potential biomarkers of diet quality is now possible using high-throughput metabolomics data, which can isolate and identify thousands of small molecules in biospecimens [10]. Consequently, its application in understanding the metabolic pathways of diet–disease associations has garnered significant interest [11]. Several investigations have examined the role of the metabolome, largely determined by dietary intake, in reflecting physiological change and chronic disease risk [12].

Although several studies have focused on diet-related metabolites of chronic disease, few have focused on age-related decline in physical function and frailty. To identify metabolites that may mediate the relationship between diet and frailty, this investigation aimed to examine the plasma metabolite biomarkers of dietary patterns, as assessed by the MDS, MIND, and AHEI indices, all of which have demonstrated associations with age-related decline in physical function and/or frailty. Additionally, we examined the mediating effect of diet-based metabolites and frailty status using data from the BLSA.

## 2. Materials and Methods

### 2.1. Study Design and Participants

The BLSA is an ongoing open cohort initiated in 1958 to study normative aging. Study protocols and detailed information on the BLSA cohort are provided elsewhere [13]. Briefly, the BLSA cohort includes community-dwelling men and women who reside primarily in the Washington DC–Baltimore area and are seen every 1–4 years from study enrollment until death. BLSA participants undergo comprehensive assessments of physical and neurocognitive function; provide information on medical history and complete dietary assessment questionnaires; and undergo laboratory and radiologic tests and other measurements at each visit during, on average, a 3-day stay at the Clinical Research Unit of the Intramural Research Program of the National Institute on Aging (IRP, NIA) or during a home visit for the most debilitated. These analyses included 806 BLSA participants who were 65 years or older and had plasma metabolomics evaluated via the Biocrates p500, a known frailty status and dietary assessment. The study protocol was approved by the National Institutes of Health Intramural Research Program Institutional Review Board, and informed consent was obtained from participants at each visit.

### 2.2. Dietary Assessment and Construction of Diet Scores

Dietary intake data were derived from a validated Food Frequency Questionnaire (FFQ) initiated at the BLSA in 2005. Initially, the FFQ was administered by clinic staff using paper forms, and beginning in 2016, the assessment was conducted using computer-based REDCap surveys. The University of Minnesota Nutrient Data System for Research program was used to generate energy and nutrient estimates. We included only valid dietary data, defined as daily energy intake between 600 and 4800 kcal [12]. Three dietary patterns were examined—the MDS, MIND, and AHEI scores (Appendix A)—and adherence scores for each diet were calculated.

*MDS score*. The MDS is based on the Greek food pyramid [3]. The MDS was calculated by assessing the intake of nine food groups and nutrients, including the consumption of beneficial food components (vegetables, legumes, fruits and nuts, whole grains, fish, and the monounsaturated-to-saturated fatty acid ratio) and detrimental food components (meat, dairy, and alcohol). Participants received a score of 1 if their consumption of beneficial components was at or above the median, their consumption of detrimental components was below the median, and their consumption of alcohol was moderate. A summary score, ranging from 0 to 9, was then calculated, with a higher score indicating better adherence to a Mediterranean-style diet.

*MIND score*. The MIND score is based on the consumption of 15 food groups: green leafy vegetables, other vegetables (excluding potatoes and potato products for these analyses), berries, nuts, olive oil, butter and margarine, cheese, whole grains, non-fried fish, beans, non-fried poultry, red meat and its products, fast/fried foods, pastries and sweets, and wine [5]. Individuals received a score of 0, 0.5, or 1 based on servings or frequency of consumption of each food group per week. Scores for the MIND pattern ranged from 0 to 15, with higher scores indicative of better diet quality.

*AHEI score*. The AHEI score includes 11 food and nutrient components previously found to be associated with chronic disease [4]. Participants received a proportionate score from 0 to 10 on the basis of their intake, for which a higher intake of vegetables, fruit, whole grains, nuts and legumes, and long-chain and polyunsaturated fatty acids; a lower intake of sugar-sweetened beverages and fruit juices, red/processed meat, *trans* fat, and sodium; and a moderate intake of alcohol contributed to higher scores. Scores for each component were then summed to derive an overall AHEI score ranging from 0 to 110, with higher scores indicating better diet quality.

The MIND and AHEI diet scores were calculated for every visit with valid energy intake data, and the averaged diet score across all visits was used for each participant. The MDS food groups were created using sample medians as cutoff points. Similar to MIND and AHEI, median values were calculated using the average values of all.

### 2.3. Frailty Index (FI)

The Frailty Index (FI) was constructed using 44 variables selected from the procedure outlined by Searle et al. [14]. The items selected included the reported difficulty with 15 basic and instrumental activities of daily living (ADL/IADL: walking up 10 steps, lifting and carrying 10 lbs, getting into and out of bed/chairs, bathing and showering, dressing, eating, using the toilet, walking across a small room, performing heavy housework, preparing one’s own meals, shopping for personal items, using the telephone, taking medication, managing finances, and urinary or fecal incontinence), self-rated health (assessed using the short-form health survey SF-12), five items from the Center for Epidemiologic Studies Depression (CES-D) scale (feeling depressed, feeling as though everything is an effort, inability to get going, feeling lonely, and feeling happy), four items of the Mini-Mental State Examination (MMSE: orientation to time, orientation to place, attention, and recall), and the prevalence of 15 common age-related conditions (cancer, anemia, hypertension, diabetes, heart disease, congestive heart failure, stroke, peripheral artery disease, COPD, chronic kidney disease, hip replacement, joint pain, depression, Parkinson’s, and cognitive impairment), 5% weight loss in the past year, low physical activity (lowest quartile of physical activity in the past year), slowness (lowest quintile walking speed stratified by sex and height), and weakness (lowest quintile grip strength stratified by sex and BMI). The cutoffs and scoring scheme are detailed in Appendix A. FI was calculated for participants with less than 20% missing data at each study visit.

### 2.4. Assessment of Plasma Metabolites

The details of the metabolomic assessment are provided elsewhere [15]. In brief, plasma specimens were obtained from overnight fasting blood samples and stored at −80 °C. Metabolites were extracted and measured using liquid chromatography–tandem mass spectrometry (LC–MS/MS) at Biocrates Life Sciences AG (Innsbruck, Austria) using the MxP^®^ Quant 500 kit. Lipids and hexoses were measured by flow injection analysis–tandem mass spectrometry (FIA-MS/MS) using a 5500 QTRAP^®^ instrument (AB Sciex, Darmstadt, Germany) with an electrospray ionization source. Twenty-six biochemical classes measured by liquid chromatography–tandem mass spectrometry (LC–MS/MS) using the same 5500 QTRAP^®^ instrument, including amino acids, related amino acids, carboxylic acids, fatty acids, indole derivatives, biogenic amines, bile acids, cresols, alkaloids, amine oxides, hormones, vitamins and cofactors, and nucleobases related metabolites, were quantified using appropriate mass spectrometry software (Sciex Analyst^®^) and imported into Biocrates MetIDQ™ software for further analysis. The accuracy of the measurements, as determined by internal calibrators, was in the normal range of the method for all analytes. Quality control samples were within the predefined tolerances of the method. Metabolites with values below the limit of detection (LOD) were set as missing, and those with >30% missing were removed from the analysis. For metabolites with ≤30% missing, values were set at half the minimum value. Of the 622 metabolites measured, 466 passed quality control (Appendix A).

### 2.5. Assessment of Covariates

Participant demographic characteristics, including age and sex, were collected during a medical examination. Body mass index (BMI) was calculated as the ratio of weight (kg) and height squared (m^2^). Weight and height were measured using a calibrated scale. Serum creatinine was measured using an isotope dilution mass spectrometry (IDMS)-traceable serum creatinine assay. The estimated glomerular filtration rate (eGFR) was calculated using the MDRD equation [16].

### 2.6. Statistical Analysis

Cross-sectional associations between each dietary pattern score (MIND, MDS, and AHEI) and individual metabolites were assessed using linear regression models, with diet scores as the dependent variables and metabolites as the independent variables. Models were adjusted for age, sex, eGFR, and total energy intake. A *p*-value of ≤0.05 (after correction for false discovery rate, FDR) was considered statistically significant.

To identify a set of metabolites that best predicts each diet score, we implemented regularized regression using *eNetXplorer* [17], an R package that tests the accuracy and significance of a family of elastic net generalized linear models ranging from ridge regression (alpha = 0) to lasso regression (alpha = 1). To select the optimal model, the elastic net mixing parameter alpha was scanned from 0 to 1 in increments of 0.1. For MIND and AHEI, we further tested values between 0 and 0.1 at increments of 0.01. For each alpha value, a grid search was performed over 100 penalty parameter λ values. For each λ, elastic net cross-validation models were generated for 500 runs, in which each run randomly assigned observations among 5 folds. The chosen regularization λ was determined by maximizing a quality function (QF) that compared the out-of-bag (i.e., not used in training) predicted response with the observed response. We used Pearson’s correlation, the default QF provided for linear regression. From the maximum QF as a function of alpha, in turn, the optimal alpha value was determined. Following a similar procedure, 125 null models were generated in the same fashion, each of them trained and tested on a randomly shuffled response vector. Significant metabolites were determined by comparing variable importance criteria—namely, coefficients (size effects) and frequencies—between the model and the null model ensemble [17]. The predicted values, or the sum of the coefficient-weighted scores for significant metabolites at the optimal alpha and λ values, were used as the metabolomic diet score. The optimal mixing parameter alpha values were 0.4, 0.02, and 0 for the MDS, MIND, and AHEI, respectively (Appendix A). The corresponding penalty parameter λ values for these models were 0.093, 0.766, and 6.936, respectively (Appendix A).

To determine whether the metabolites mediate the relationship between diet and FI, we conducted mediation analyses using the *mediation* R package [18]. This was performed in two steps. First, a mediation model was built in which the mediator (metabolomic diet score) was regressed on the diet score. Next, FI was regressed on the diet score adjusted for the metabolomic diet score. Four estimates were then calculated: the total effect (TE), which reflected the effect of diet on FI without consideration of the metabolomic diet score; the average direct effect (ADE) and the average causal mediated effect (ACME), which reflected the direct and mediated/indirect effects of diet on FI, respectively; and the proportion of mediation that captures the ratio between ACME and TE. The average estimates and 95% CI were calculated from 500 non-parametric bootstrapped samples. A value of *p* ≤ 0.05 was considered statistically significant for mediation. All analyses were conducted using R version 3.6.2.

## 3. Results

### 3.1. Association of Dietary Patterns with Frailty Index

The demographic, clinical, and dietary characteristics of 806 BLSA participants are presented in Table 1. The average scores for the MDS, MIND, and AHEI were 4.3, 8.04, and 55.2, respectively. The correlations between the three dietary scores from highest to lowest were as follows: MIND and AHEI (r = 0.66), MIND and MDS (r = 0.64), and MDS and AHEI (r = 0.57). All dietary scores were inversely associated with FI (AHEI: β = −0.008 ± 0.002, *p* < 0.001; MDS: β = −0.006 ± 0.002, *p* < 0.001; MIND: β = −0.006 ± 0.002, *p* = 0.005).

### 3.2. Metabolomic Biomarkers of Dietary Patterns

Of the 466 metabolites assessed, we found 236, 218, and 278 metabolites associated with the MDS, MIND, and AHEI, respectively, after correction for false discovery rate (Appendix A). There were 176 metabolites associated with all three diet scores. Of the 191 triglyceride metabolites measured, 123 were associated with all diet scores. These triglycerides represented two clusters of highly correlated metabolites (Appendix A). The first cluster of six triglycerides was positively correlated with diet scores, while the second cluster of 117 metabolites was negatively associated with diet scores. The most significant positively associated triglyceride was TG(22:6_34:3) for all diet scores. The most significant negatively associated triglycerides were TG(22:4_34:2) (β = −0.21 ± 0.036, *p* = 4.15 × 10^−7^), TG(16:0_35:2) (β = −0.23 ± 0.036, *p* = 2.42 × 10^−8^), and TG(20:4_32:0) (β = 0.29 ± 0.034, *p* = 2.96 × 10^−14^) for MIND, MDS, and AHEI, respectively (Figure 1). There were 24 diet-associated phosphatidylcholine (PC) metabolites. One highly correlated cluster of eight metabolites was negatively associated with diet scores, with PC diacyl C40:4 being the most significant (Appendix A and Figure 1). There were two clusters of 16 PC metabolites positively associated with diet scores, with the most significant being PC diacyl C38:6 (β = 0.24 ± 0.035, *p* = 7.04 × 10^−9^), PC diacyl C42:2 (β = 0.24 ± 0.035, *p* = 8.42 × 10^−9^), and PC diacyl C42:0 (β = 0.21 ± 0.035, *p* = 2.54 × 10^−8^) for MIND, MDS, and AHEI, respectively. The 29 remaining metabolites were from the classes of cholesterol esters (5), ceramides (4), diglycerides (3), lysophosphatidylcholines (3), sphingomyelins (3), amino acids and amino acid-related compounds (3), bile acids (2), fatty acids (2), indoles and derivatives (2), acylcarnitines (1), and carboxylic acids (1).

### 3.3. Metabolomic Signature Mediate Associations between Diet and Frailty Index

To create a metabolomic signature of diet, significant metabolites were selected using elastic net regression. There were 10, 41, and 59 metabolites selected using elastic net regression for the MDS, MIND, and AHEI, respectively (Figure 2). Four metabolites (tryptophan betaine (TrpBetaine), CE (17:1), PCaaC40:5, and PCaeC42:3) were included in all three dietary pattern scores. The correlations between the metabolomic diet score (metDS) and the observed diet score were 0.42, 0.49, and 0.56 for the MDS, MIND, and AHEI, respectively (Figure 2). Adjusting for covariates (sex, age, eGFR, and total energy intake), the metDS explained 23.4%, 26.9%, and 31.7% of the variance in the MDS, MIND, and AHEI, respectively. To test whether the metDS mediated the association between the diet indices and the frailty index, we conducted mediation analyses (Table 2). For MIND and AHEI, the indirect effect (or average causal mediation effect) was significant (*p* < 0.001), supporting the mediating effect of the metDS on diet and FI. The metDS mediated 55% and 61% of the association between MIND and AHEI and FI, respectively.

## 4. Discussion

We identified the plasma metabolomic profile of three different summary dietary pattern measures in 806 BLSA participants aged 65 years or older. We found over 200 plasma metabolites associated with each dietary pattern, with 176 metabolites in common. To create a metabolomic signature of diet (metDS), significant metabolites were selected using regularized regression methods. The correlation between the metDS and the measured dietary patterns was consistent with previous reports [19]. There were four metabolites (TrpBetaine, CE(17:1), PCaa C40:5, and PCae C42:3) that were common to all three diet pattern summary scores. The identified metabolomic signatures represent a wide range of lipids species, amino acids and derivatives, bile acids, and vitamins. This metabolomic signature partially mediated the relationship between diet and frailty index for MIND and AHEI. This suggests that adherence to the dietary patterns examined may slow the accumulation of deficits through the modulation of the different pathways represented by the various metabolites. Understanding the roles of the diet-related metabolites provides us with clues about the underlying molecular mechanisms of the effects of diet on frailty status. The potential mechanisms are described below.

There have been several metabolomic investigations of dietary patterns in observational studies [20]. A key challenge in comparing metabolomic studies of diet lies in the different methodologies used to assess both diet and metabolites, as coverage of metabolites differs across platforms with little overlap [20]. In this regard, the Fenland study, a large cohort of 10,806 participants, used the MDS to assess diet and the Biocrates platform to assess 175 plasma metabolites [19]. They identified 66 diet-associated metabolites, of which 54 passed quality control in the BLSA, and 37, or 69% were significantly associated with MDS. In particular, there was a high degree of agreement for the phosphatidylcholines (PC), lysophosphatidylcholines (LPC), and sphingomyelins [19]. This indicates that when using the same assessment tools to measure metabolites and diet, there is a high degree of consistency in diet-associated metabolites across geographically distinct populations. Since the average age of Fenland study participants was much lower than that of the BLSA participants (48.4 vs. 73.3 years), this suggests that diet biomarkers are consistent, independent of age. Interestingly, a recent study reported that adherence to the World Cancer Research Fund dietary recommendations was associated with a lower abundance of 10 phosphatidylcholines in a study of 195 participants with colorectal cancer [19], most of which were consistent with the PCs identified with MDS. Together, these studies suggest that PCs may be an indicator of habitual adherence to a healthy diet.

Previous observational and intervention studies have found the MDS and AHEI to be associated with favorable lipid profiles, including lower triglycerides [21,22]. Triglycerides were the most represented metabolite in the targeted metabolomics platform, and they were the class with the greatest number of metabolites associated with diet scores in this study. The significant triglyceride metabolites reflected two correlated clusters. Interestingly, the positively associated triglycerides contain eicosapentaenoic acid (EPA) or docosahexaenoic acid (DHA), with five of the triglycerides containing DHA or EPA and the sixth triglyceride, TG (18:1_38:7), possibly containing DHA or EPA. A positive association with EPA and DHA was also observed in free fatty acids and cholesterol esters in our data. Food sources of DHA and EPA include nuts, seeds, and fish; thus, these positive associations are consistent with the consumption of these source foods, which is considered consistent with a healthy dietary pattern. The negatively associated TG metabolites with diet scores include a large number of fatty acid species, including arachidonic acid, linoleic acid, palmitic acid, and myristic acid. These data suggest that adherence to a healthier dietary pattern is associated with longer-chain fatty acids; however, a more detailed lipidomic analysis that can identify all side chains may provide a more comprehensive TG profile of dietary intake.

In addition to TG and PC, several other lipid metabolites were associated with dietary patterns, including sphingolipids, such as ceramides and sphingomyelins (SMs). Both ceramides and sphingomyelins play a structural role in cell membranes and regulate signaling as second messenger molecules [23]. These bioactive lipids have been linked with different biological processes, including apoptosis, inflammation, oxidative stress, and senescence [24,25,26]. Sphingolipids have been associated with longevity, AD, and irritable bowel syndrome [27,28,29]. In the BLSA, ceramides (d18:1/18:0 and d18:1/20:0) and SM C20:2 were negatively associated with adherence to all three dietary patterns, while the very-long-chain SM C24:1 and SM C26:1 were positively associated with adherence to all the three dietary patterns. Several studies support these observations. For example, in the Framingham offspring study, adherence to the MDS was inversely associated with d18:1/16:0 and d18:1/22:0 [30]; while d18:1/16:0 was not significant in the MDS in our study, d18:1/22:0 was trended toward an inverse association in the MDS. The negative association for Cer d18:1/22:0 was also observed in the Prudent diet in the Women’s Health Initiative (WHI) [31]. In the WHI, SM24:1 was negatively associated with a Western-style diet and positively associated with the Prudent diet, which is consistent with the positive association observed in the BLSA. Furthermore, Hex3Cer (d18:1/24:1) and Hex2Cer (d18:1/24:1) were also positively associated with all diet summaries in our study. Nervonic acid (24:1) is available as a dietary supplement and in whole foods, including swordfish, salmon, sesame seeds, and quinoa. While nervonic acid levels could be influenced by non-food sources, it is possible that the positive associations with the dietary patterns observed in the BLSA reflect the consumption of foods rich in nervonic acid. Overall, the consistent associations between lipids and different dietary patterns suggest that the lipidomic profile could be an informative biomarker of habitual diet intake.

Three amino acid derivatives associated with diet patterns were previously identified as biomarkers of specific food groups. Tryptophan betaine (TrpBetaine) is a derivative of the amino acid tryptophan that has previously been positively associated with nut intake [32,33]. Trans-4-hydroxyproline (t4-OH-Pro), which is a derivative of proline, has been positively associated with meat consumption [33]. Consistent with these observations, TrpBetaine was positively associated and t4-OH-Pro was negatively associated with adherence to diet patterns examined in these analyses. TrpBetaine, or hypaphorine, is a derivative of tryptophan that was shown to have anti-inflammatory properties through the regulation of the PI3K/Akt/mTOR signaling pathway [34]. Interestingly, TrpBetaine was included in the signature of all three dietary patterns, supporting the ability of the anti-inflammatory effect of diet to reduce the burden of frailty. Another amino acid derivative, hippuric acid (HA), was positively associated with adherence to the three dietary patterns. HA is produced by bacteria from plant phenols, and urinary HA increases with the consumption of fruits and vegetables [35]. Interestingly, there has been increasing interest in HA as a potential biomarker of frailty that could be used to monitor the development of geriatric syndromes [36]. Moreover, a recent study demonstrated that fruit and vegetable consumption mediated the association between HA and frailty [37]. Our data support these previous findings, as adherence to a “healthier” dietary pattern represents a diet rich in phenolic compounds, including fruits and vegetables. Taken together, HA could be used as a tool to identify individuals with poor diets who could be targeted for dietary intervention to improve aging trajectories.

Several metabolites associated with the dietary patterns were microbiome-derived. In particular, several tryptophan metabolites were identified. Tryptophan, present in animal protein, is utilized by gut bacteria and converted to indole, which is then absorbed by gut villi and enters the portal system of the liver, where it is sulfated and metabolized into indoxyl sulfate (Ind-SO4) [38]. Ind-SO4 is a metabolite characteristic of the consumption of a Western-style diet and is associated with a higher intake of animal protein [39]. Consistent with this observation, greater adherence to the dietary patterns examined in this study was inversely associated with Ind-SO4. Plasma Ind-SO4 has been associated with incident cardiovascular events in patients with mild chronic heart failure [40] and chronic kidney disease [41]. Dietary patterns play an important role in the composition of the gut microbiome and the bioactive compounds they produce. In a small pilot study, 10 healthy participants were randomized to a fast-food diet and a Mediterranean-style diet for a period of 4 days, and the effects of these diets on the gut microbiome composition and plasma concentrations of bacteria-produced metabolites were examined. Of note, the researchers observed that metabolites such as indole-3-propionic acid (3-IPA) and indole-3 acetic acid (3-IAA) increased in participants on the Mediterranean diet and decreased in those on the fast-food diet [42]. In our analyses of the BLSA, 3-IPA was positively associated with adherence to the three dietary patterns, consistent with the observational study. Unexpectedly, 3-IAA was negatively associated with adherence to the MDS and AHEI. Both 3-IAA and 3-IPA have shown antioxidant and low pro-oxidative properties [43,44] and immune regulation through the activation of the aryl hydrocarbon receptor [45,46], and 3-IPA has been inversely associated with diabetes and inflammation [47]. The divergent association between IPA and IAA may have resulted from microbial changes, as dietary patterns have been shown to change gut microbiota composition [48]. IAA is predominantly produced by *Bacteroides* [47,49,50], and IPA is produced predominantly by *Clostridium sporogenes* [50,51]. Future studies should investigate the effect of dietary patterns on the gut microbiome and downstream effects on plasma metabolites to fully understand the results that were observed in this study.

Our study has several strengths, including dietary assessment using a validated FFQ and the study of dietary patterns, which is preferred over studying individual foods and nutrients. Moreover, our investigation included a comparison of different dietary patterns, which is a significant strength. Our study outcome, frailty, was characterized using a validated frailty index, which took deficits in health into account. This study also has several limitations. First, it is a cross-sectional study, and thus the results we report are associations and we cannot infer causal relationships. Second, the participants of the BLSA are more educated and healthier than the general US population of comparable age, and thus the generalizability of the study findings may be limited. However, the consistency of the results with studies on populations in other geographic regions with different health statuses suggests that the results are largely generalizable. Finally, although the metabolite panel assessed in this study was sizeable, we utilized a targeted metabolomics approach; therefore, it is likely that other relevant diet-related metabolites were not measured in our study. Therefore, it is important that future studies expand upon the work conducted here and by other investigators to build a comprehensive catalog of diet-related metabolites that could be used to monitor individuals’ dietary habits.

## 5. Conclusions

In summary, this study characterized the plasma metabolomics profile of adherence to three dietary patterns and found that large proportions of diet-associated metabolites were shared across three dietary pattern assessments, supporting common underlying mechanisms of the positive associations between better health and function and adherence to these dietary patterns. Importantly, evaluating plasma metabolites could be a useful tool to track adherence to a high-quality diet and promote healthy aging.

## Figures and Tables

**Figure 1 nutrients-14-02237-f001:**
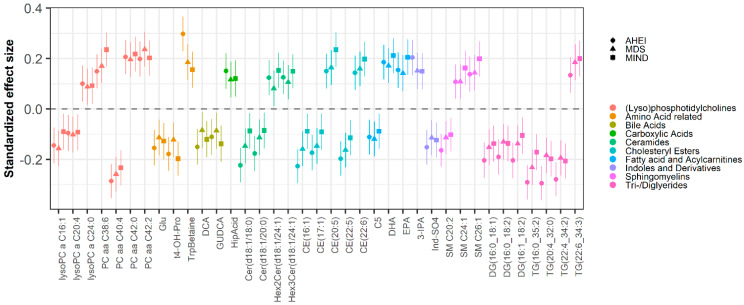
Association of plasma metabolites with Mediterranean diet score (MDS; triangle), Mediterranean–DASH Intervention for Neurodegenerative Delay (MIND) score (square), and Alternative Healthy Eating Index (AHEI) score (circle). The figure represents standardized coefficients from the regression model in which each metabolite is regressed on dietary patterns.

**Figure 2 nutrients-14-02237-f002:**
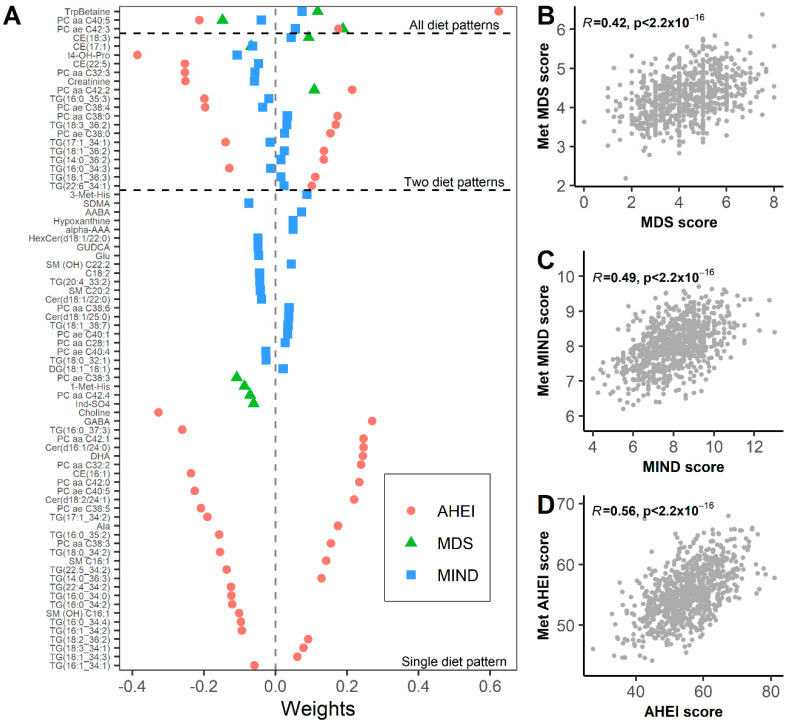
Metabolites selected to create metabolomic diet score. (**A**) Using elastic net regression, 10, 41, and 59 metabolites were selected to create a metabolomic diet score (metDS) for MDS, MIND, and AHEI, respectively. There were 4 metabolites common to all three dietary patterns, 19 metabolites associated with two dietary patterns, and 60 metabolites unique to individual dietary pattern score. The metDS showed moderate associations with the measured diet scores, with a correlation of 0.42 for MDS (**B**), 0.49 for MIND (**C**), and 0.56 for AHEI (**D**).

**Table 1 nutrients-14-02237-t001:** Socio-demographic characteristics, dietary scores, and frailty index components.

	Value * (*n* = 806)
Age, years	73.3 (7.1, 65–95)
Men, *n* (%)	391 (48.5)
Race, *n* (%)	
White	577 (71.6)
Black	183 (22.7)
Other	46 (5.7)
MIND score	8.0 (1.4, 4–13)
MDS score	4.3 (1.3, 0–8)
AHEI score	55.2 (8.1, 27.2–81.1)
eGFR, mL/min/1.73 m^2^	69.1 (15.4, 15.3–127.2)
Frailty Index	0.1 (0.1, 0–0.4)

* Values are expressed as means (SD, minimum–maximum) for continuous variables and *n* (%) for categorical variables.

**Table 2 nutrients-14-02237-t002:** Results of mediation analysis of metabolomic diet score on the association between diet and frailty index.

	MDS			MIND			AHEI		
	Estimate	95% CI	*p*	Estimate	95% CI	*p*	Estimate	95% CI	*p*
ACME	−0.001	(−0.003, −0.00001)	0.076	−0.003	(−0.004, −0.001)	<0.001	−0.0005	(−0.0008, −0.0003)	<0.001
ADE	−0.003	(−0.006, 0.0004)	0.052	−0.002	(−0.005, 0.001)	0.256	−0.0004	(−0.001, 0.0002)	0.14
TE	−0.004	(−0.007, −0.001)	0.012	−0.005	(−0.008, −0.001)	0.004	−0.001	(−0.0015, −0.0004)	<0.001
PM	0.29	(−0.05, 0.97)	0.088	0.55	(0.21, 0.95)	0.004	0.61	(0.26, 1.31)	<0.001

ACME—average causal mediation effect; ADE—average direct effect; TE—total effect; PM—proportion mediation (ratio of ACME to TE).

## Data Availability

Data from the Baltimore Longitudinal Study of Aging are available through the submission of the research proposal through https://www.blsa.nih.gov/ (accessed on 20 March 2022).

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
