# Peer review of "Metabolomic Profile of Different Dietary Patterns and Their Association with Frailty Index in Community-Dwelling Older Men and Women"

_nutrients, 2022, doi:10.3390/nu14112237_

Round 1

Reviewer 1 Report

The authors illustrate a very meaningful theme. Diet quality has been associated with slower rates of aging, and the authors have made appropriate efforts to discover plasma metabolomic biomarkers of dietary patterns and explored whether these 16 metabolites mediate the relationship between diet and healthy aging as assessed by frailty index 17 (FI) in 806 participants of the Baltimore Longitudinal Study of Aging. The manuscript is well-written and explanatory. Method and Results are well-defined. Statistical analyses are adequate. The language is understandable with no grammatical and syntaxerrors. The references are up-to-date. 

1, The abbreviation, first appearing in the manuscript, shall indicate its full name.

2, Authors should strongly justify the necessity to conduct the described research. This part of the introduction is insufficient. Both, in the introduction and in the discussion, the following paragraphs are often thematically unrelated. Authors should take care of the quality of the text.

3, The study protocol was approved by the National Institutes of Health Intramural Research Program Institutional Review Board and informed consent was obtained from participants at each visit. Please provide the approval number.

Author Response

Reviewer 1.

The authors illustrate a very meaningful theme. Diet quality has been associated with slower rates of aging, and the authors have made appropriate efforts to discover plasma metabolomic biomarkers of dietary patterns and explored whether these 16 metabolites mediate the relationship between diet and healthy aging as assessed by frailty index 17 (FI) in 806 participants of the Baltimore Longitudinal Study of Aging. The manuscript is well-written and explanatory. Method and Results are well-defined. Statistical analyses are adequate. The language is understandable with no grammatical and syntaxerrors. The references are up-to-date. 

1, The abbreviation, first appearing in the manuscript, shall indicate its full name.

>>We identified two abbreviation that was indicated in full on line 321 for EPA and DHA. This was corrected.

2, Authors should strongly justify the necessity to conduct the described research. This part of the introduction is insufficient. Both, in the introduction and in the discussion, the following paragraphs are often thematically unrelated. Authors should take care of the quality of the text.

>>Thank you for the comment, we modified a section in the introduction:

Line (77): To identify metabolites that may mediate the relationship between diet and frailty, this investigation aims to examine the plasma metabolite biomarkers of dietary patterns as assessed by the MDS, the MIND and the AHEI indices all of which have demonstrated associations with age-related decline in physical function and/or frailty

3, The study protocol was approved by the National Institutes of Health Intramural Research Program Institutional Review Board and informed consent was obtained from participants at each visit. Please provide the approval number.

>>We have included the approval date: (03-AG-0325; Approval date: 10/5/2021)

Reviewer 2 Report

In manuscript “Metabolomic profile of different dietary patterns and their association with frailty index in community-dwelling older men and women”, the authors measured the plasma metabolite biomarkers of dietary patterns. The adherence to different dietary patterns was evaluated using the MDS, MIND score, and AHEI. They found that although the effect of diet is complex and not mediated by any specific biomarker, the metabolites could serve as biomarkers for poor diet quality to identify individuals for targeted dietary interventions. These findings are interesting and significant. The manuscript is well written and experimental data are presented in a logical way. 

Author Response

thank you for your comments

Reviewer 3 Report

The article “Metabolomic profile of different dietary patterns and their association with frailty index in community-dwelling older men and women ” is overall well written and readable.  The results are clear and of benefit for clinical practice. 

I have no further comments to this manuscript.

Author Response

Thank you, we appreciate your generous comments.